# ModReduce: A Multi-Knowledge Distillation Framework with Online Learning

## Abstract

Deep neural networks have produced revolutionary results in many applications; however, the computational resources required to use such models are expensive in terms of processing power and memory space. Research has been conducted in the field of knowledge distillation, aiming to enhance the performance of smaller models. Knowledge distillation transfers knowledge from large networks into smaller ones. Literature defines three types of knowledge that can be transferred: response-based, relational-based, and feature-based. To the best of our knowledge, only transferring one or two types of knowledge has been studied before, but transferring all three remains unexplored. In this paper, we propose ModReduce, a framework designed to transfer the three knowledge types in a unified manner using a combination of offline and online knowledge distillation. Moreover, an extensive experimental study on the effects of combining different knowledge types on student models' generalization and overall performance has been performed. Our experiments showed that ModReduce outperforms state-of-the-art knowledge distillation methods in terms of Average Relative Improvement.

## 1 Introduction

The term knowledge distillation was formally popularized in the work of Hinton et al. (2015), and it refers to transferring knowledge from a large pre-trained model to a smaller one, aiming to retain comparable performance to the large model. Knowledge distillation has been receiving increasing attention from the research community due to its promising results. The methods by which knowledge can be distilled vary widely based on several factors like knowledge type, the distillation algorithm, and the teacher-student architecture (Gou et al., 2021). Response-based knowledge uses the large model's logits as the teacher model's knowledge. The main idea is that the student optimizes its training over the soft targets, or the softened probability distribution, produced by the teacher model instead of using discrete labels (Hinton et al., 2015). While this method showed great success, one of its major drawbacks is that it disregards the knowledge a teacher model retains in its intermediate layers. This encouraged researchers to introduce methods that capture the knowledge in the intermediate layers of the teacher model, feature knowledge. Feature-based algorithms focus on the features of the teacher model's intermediate layers to guide the student's learning. The challenge is that the teacher and the student models have different abstraction levels, which makes it one of the objectives of the distillation process to determine the best layer associations for maximum performance (Tung & Mori, 2019; Passalis et al., 2020; Kornblith et al., 2019). Relational-based methods focus on the relationships between different data instances and different activations and neurons (Gou et al., 2021). Several algorithms and methods have been introduced, focusing on distilling one or two of the knowledge sources from a teacher model to a student model. While they show promising results, no research addresses the issue of distilling the three discussed knowledge types instead of only one or two. To the best of our knowledge, ModReduce is the first work to explore this area. Moreover, we explore combining offline and online distillation strategies to achieve this goal. For online learning distillation, we have explored four different techniques for knowledge aggregation: Peer Collaborative Learning (PCL) (Wu & Gong, 2021), On-the-fly Native Ensembling (ONE) (Zhu et al., 2018), Fully Connected Layers (FC), and Weighted Averaging.

ModReduce exhibits the following characteristics:

1. Agnostic of the offline distillation algorithm used. The platform supports different implementations for each knowledge category distillation, including state-of-the-art algorithms.
2. Capable of executing different online learning algorithms.
3. Agnostic of the teacher and student model architectures.

Moreover, we introduce a new benchmark which is a union of the benchmarks available at Chen et al. (2021); Tian et al. (2019). With this broad benchmark, several findings have been concluded.

## 2 RELATED WORK

Before digging deeper into different knowledge distillation schemes, it is better to explore the different knowledge representations a neural network possesses.

### 2.1 KNOWLEDGE SOURCES

#### 2.1.1 RESPONSE-BASED

Response-based knowledge is defined as the response of a neural network whenever it is presented with a particular input. The most popular response-based knowledge distillation scheme in image classification tasks is using "soft targets," where the aforementioned probability distribution is controlled using a temperature factor, $T$, which is used to soften the probability distribution. According to Hinton et al. (2015), these soft targets provide informative dark knowledge from the teacher model. The soft targets "Hinton" approach is the most popular approach with the best results thus far utilizing this type of knowledge. Unfortunately, this kind of knowledge is blind to the inner features in the hidden layers of the model, as it only focuses on the final outputs.

#### 2.1.2 FEATURE-BASED

Feature-based knowledge extends on the idea of response-based knowledge and takes the outputs of intermediate layers into consideration. Accounting for the outputs of the intermediate layers is important because deep neural networks can learn features with different levels of abstraction. For instance, a deep CNN can learn abstract features like straight and curved lines in the shallowest layers while detecting features with higher complexity at the deeper layers (Bengio et al., 2013). This idea is useful for constructing teacher-student architectures since this type of knowledge can be used in the training of the student network. Many knowledge distillation techniques use a distillation loss function that accounts for feature-based knowledge. Equation 1 represents a general form of the distillation loss function for feature-based knowledge, where $f_t(x)$ and $f_s(x)$ are the feature maps of the teacher and student models respectively; $\phi_t$ and $\phi_s$ represent the transformation function of the teacher and student models feature maps and $l_f(.)$ is a similarity measure between the feature maps.

$$L_{FeaD}((f_t(x), f_s(x)) = l_f(\phi_t(f_t(x)), \phi_s(f_s(x))) \tag{1}$$

The state-of-the-art framework in feature knowledge distillation was introduced in Chen et al. (2021), which aimed to match the semantics between the teacher and student. They introduced semantic calibration for cross-layer knowledge distillation that made better use of the intermediate knowledge by matching the semantic level of the transferred knowledge. Then they used an attention mechanism to automatically learn a soft layer association with multiple targets, which helped the student model in learning from multiple semantically matched hidden layers instead of just one fixed layer.

#### 2.1.3 RELATION-BASED

While the response-based distillation captures the knowledge in the output layers of the teacher model, and feature-based distillation captures knowledge contained in the intermediate layers, the relational-based distillation captures the interrelations between training data examples. Several techniques have been proposed to capture the relations between the training data. Instance Relationship Graph (IRG) (Liu et al., 2019) introduced a methodology of knowledge distillation based on constructing a graph where features are represented as vertices and relations as edges. Relational Knowledge Distillation (RKD) (Park et al., 2019) proposed measuring the relations between training data

instances using distance-wise and angle-wise losses that penalize structural differences in relations. Contrastive Representation Distillation (CRD), which is the state-of-the-art in relational knowledge distillation (Tian et al., 2019), captures important structural knowledge of the teacher network. It trains a student to capture significantly more information in the teacher's representation of the data using objective contrastive learning, which encourages the student to map similar inputs to close representations, in some metric space, while mapping different inputs to distant representations.

## 2.2 DISTILLATION SCHEMES

In this section, we overview the existing distillation schemes in the literature, which can be divided into three schemes: offline distillation, online distillation, and self-learning distillation. We focus on the first two schemes since they are more relevant to our target architecture.

### 2.2.1 OFFLINE DISTILLATION

This is the most basic and popular kind of distillation. It was introduced alongside the concept of distillation by Hinton et al. (2015). This scheme transfers knowledge from a pre-trained expert teacher model into a student model. The whole training process takes place in two phases; first, the training of the teacher model on the set of training samples before distillation. The second phase is extracting knowledge from the teacher model and passing it to the student model. The knowledge is extracted from features, responses, or relations as mentioned in section 2.1. Offline distillation is simple and straightforward to implement as it employs one-way knowledge transfer from a trained teacher. In addition, the student model is usually smaller in size and simpler to train. On the other hand, in this scheme, the model capacity gap always exists and cannot be avoided due to the difference in complexity between the teacher and student models. *Model capacity* could be defined as a measure of a DNN size based on the number of nodes and layers. The offline distillation methods focus on improving knowledge transfer in several aspects. One aspect is regarding the design of the student model and the knowledge type. For instance, in Romero et al. (2014), the student model is deeper than the teacher model but it is much thinner at the same time. Hints from the inner hidden layers of the teacher model are taught to the student model to guide the training process. Another aspect is the loss functions for matching features or distributions matching.

### 2.2.2 ONLINE DISTILLATION

In the online distillation scheme, both the teacher and student models are being trained and updated simultaneously. Online learning has been proven to improve the generalization ability of a network by training it simultaneously with a pool of other networks. Moreover, online learning supports heterogeneity in student networks as they can vary in architecture and size. Several techniques have been proposed for the online learning scheme, such as Peer Collaborative Learning (PCL), On-the-fly Native Ensemble (ONE), and weighted averaging. Peer Collaborative Learning (Wu & Gong, 2021) integrates online ensembling and network collaboration into a unified framework. PCL constructs a multi-branch network for training, in which each branch is called a peer. Random augmentation multiple times is performed on the inputs to peers and assemble feature representations outputted from peers with an additional classifier as the peer ensemble teacher. Moreover, PCL employs the temporal mean model of each peer as the peer mean teacher to collaboratively transfer knowledge among peers, which helps each peer to learn richer knowledge and facilitates optimizing a more stable model with better generalization. In ONE (Zhu et al., 2018), training is only a single multi-branch network while simultaneously establishing a strong teacher on-the-fly to enhance the learning of the target network. The auxiliary branches share the low-level layers with the target network, with each branch, together with the shared layers, acting as an individual model. The ensemble of those branches builds the teacher model. The training is performed in a closed loop fashion where the teacher aggregates knowledge from branch models on-the-fly, and this knowledge is distilled back to the branches to enhance the models' learning. Evaluations of ONE report enhancement of the generalization performance while maintaining the computational efficiency.

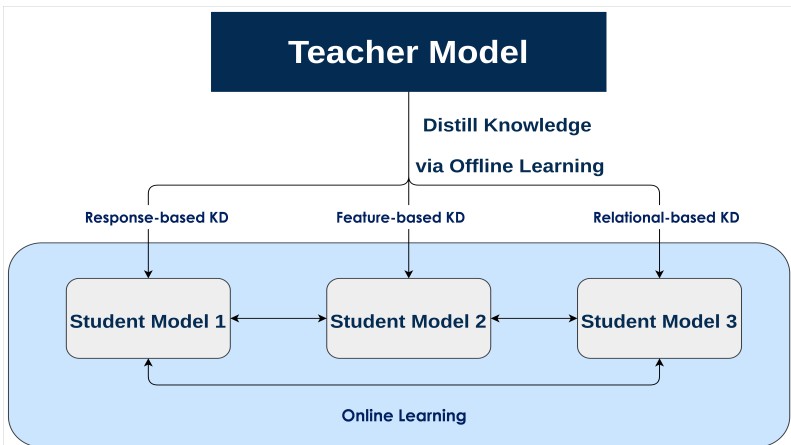

Figure 1: ModReduce Architecture

# 3 MODREDUCE

## 3.1 FRAMEWORK STRUCTURE

The structure of our framework is as follows, one cumbersome teacher model that is already pre-trained and has high accuracy in the specific task and three identical untrained student models. These student models are smaller in size than the teacher model. Notably, the student models under the standard training process obtain lower accuracy than the teacher model. The framework operation is divided into two phases, the offline distillation phase, and the online distillation phase. Both these phases are performed consecutively at each training step. First, offline distillation is performed between the teacher model and each student model separately, using different knowledge sources for each student. Then, online distillation is performed among the student models to aggregate the knowledge gained in the offline phase. For online learning, we implemented different techniques as mentioned in section 2.2.2.

## 3.2 OFFLINE TRAINING SETUP

This section will go through the functionality and loss calculation of each offline distillation method used in our algorithm. We used Hinton for response distillation, SemCKD for feature distillation, and CRD for relational distillation.

### 3.2.1 HINTON

By "Hinton" here we refer to the vanilla KD, which depends on the loss calculated between the logit layers of the teacher and student models. This method uses the outputs of the final Softmax layer of the teacher, which contains the probabilities for each class (in a classification task), then applies a temperature for these probabilities to convert them into the soft targets. Equation 2 shows how to obtain a soft target $q_i$, where $T$ is the softening temperature, $i$ is the index of the class, $z_i$ is the logit computed for class $i$, and $q_i$ is the probability of class $i$. If $T$ is greater than 1, we obtain $q_i$ values that are softened probabilities (i.e., soft targets).

$$q_i = \frac{e^{\frac{z_i}{T}}}{\sum_j e^{\frac{z_j}{T}}} \tag{2}$$

### 3.2.2 SEMCKD

As a feature knowledge distillation method, SemCKD is concerned with transferring knowledge from the intermediate layers of the teacher to the students. Moreover, it employs an attention mechanism to solve the problem of semantic mismatch caused by the difference in teacher and student

architectures which could lead to a degradation in performance. The attention mechanism automatically assigns layers from the teacher model for student layers to learn from. In addition to the attention mechanism, each student layer learns from multiple layers in the teacher model to add cross-layer supervision (Chen et al., 2021).

### 3.2.3 CRD

As for relational, or, structural knowledge, CRD is the current state-of-the-art. The original response-based knowledge transfer proposed by Hinton et al. (2015) ignores the complex inter-dependencies between the data instances, a problem CRD tries to solve by leveraging a contrastive objective to capture this output correlation (Tian et al., 2019).

### 3.3 ONLINE TRAINING SETUP

The online learning phase enhances the generalization of the student models by sharing the knowledge gained by the other models in the cohort during the offline phase. We have explored four different online learning techniques inspired from different sources to find the best one for our goal. These four techniques are PCL, ONE, FC, and Weighted Averaging.

### 3.3.1 PCL

This technique was inspired by Wu & Gong (2021). In it, the students try to learn collaboratively from each other by employing a temporal mean model copy as a representation for each student. In the online learning phase, each student tries to mimic the soft logits of the temporal mean models of its peers.

### 3.3.2 ONE

In this technique, inspired by Zhu et al. (2018), inputs and predictions are used to learn a weight for each student. Those weights are used to produce a group output from the individual student predictions, which is then used as a guide for the students to mimic.

### 3.3.3 FC

Similar to ONE, this technique tries to produce a better group output from individual student predictions for the students to follow. However, this objective is achieved here by employing a dense layer to aggregate the three individual predictions into a single one.

### 3.3.4 WEIGHTED AVERAGING

This technique has the same objective as ONE and FC, sharing knowledge between the students by aggregating their predictions into a group output that each student is penalized against. As its name suggests, we here try to learn a weight for each student, with the weights being from 0 to 1 and having a total sum of 1. Such goal is achieved by having a learnable weight for each student, and those weights are optimized based on students' performance.

### 3.4 ALGORITHM

Algorithm 1 shows ModReduce training detailed procedure.

## 4 EXPERIMENTAL SETUP

To verify our proposed hypothesis and demonstrate the effectiveness of our novel framework, we designed a flow for our experimentation. The flow chart in figure 2 describes the entire rationale of each stage.

We started with our original hypothesis question: "Does aggregating three knowledge sources in one loss function yield better results than distilling only one knowledge source?". If the answer was yes, then this is a definitive answer to the hypothesis that offline aggregation of the three knowledge

---

**Algorithm 1** ModReduce Algorithm

---

1: Load pretrained TeacherModel
2: S = {Response Distillation Model, Relational Distillation Model, Feature Distillation Model}
3: Create StudentModel[x] $\forall \{x \in S\}$
4: **for** $epoch\ in\ epochs$ **do**
5:    **for** $batch\ in\ train\_data$ **do**
6:       t_predictions = TeacherModel.predict(batch)
7:       **for** $x \in S$ **do**
8:          StudentModel[x].predictions = StudentModel[x].predict(batch)
9:          StudentModel[x].offline_loss = StudentModel[x].compute_offline_loss()
10:       **end for**
11:       group_out = calculate_group_output(StudentModel[x].predictions: $\forall x \in S$)
12:       **for** $x \in S$ **do**
13:          StudentModel[x].online_loss    =    calc_online_loss(StudentModel[x].predictions, group_out)
14:          StudentModel[x].total_loss = online_loss_weight * StudentModel[x].online_loss + offline_loss_weight * StudentModel[x].compute_offline_loss()
15:       **end for**
16:    **end for**
17: **end for**

---

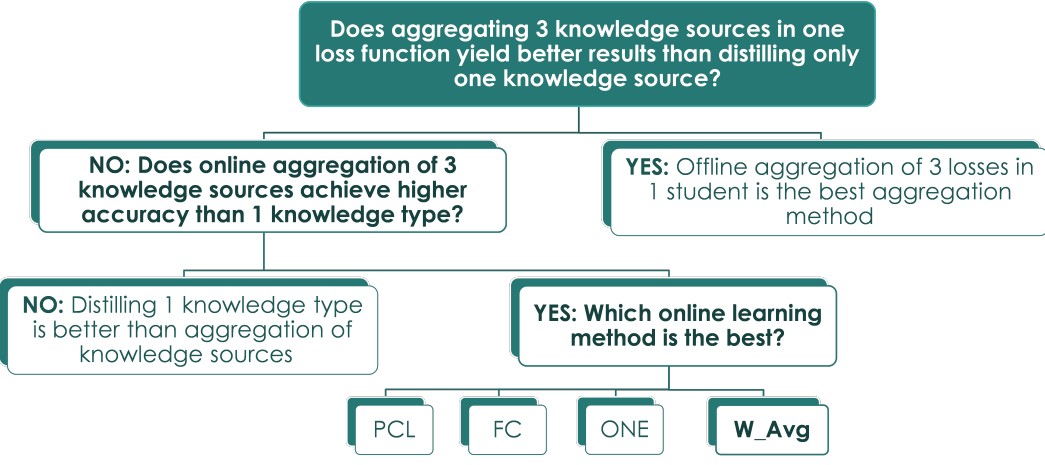

Figure 2: Experimental Flow

sources is sufficient and produces better results than distilling only one knowledge type. With that established, we would not need any further experimentation. If the answer was no, then we need to explore whether changing the aggregation method would enhance performance. So, the question now becomes: "Does online aggregation of three knowledge sources achieve better results than only one knowledge source?" If the answer to this question was no, then that would be a definitive answer to falsify our original hypothesis and clearly show that distilling only one knowledge type is better than distilling three knowledge types. However, if the answer was yes, this proves our proposed hypothesis (that aggregating three knowledge sources using online learning shall produce better results than only one knowledge source). After ensuring that the obtained results using online aggregation of three knowledge sources are higher than the state-of-the-art benchmarks, we can further investigate which online learning technique achieves the highest performance. Addressing this question, we integrated four different online learning algorithms to test the performance of our model. We experimented with Peer Collaborative Learning (PCL), Fully Connected layers (FC), On-the-fly Native Ensemble (ONE), and Weighted Averaging.

The state-of-the-art methods we compared against are Hinton (in response-based knowledge distillation), SemCKD (in feature-based knowledge distillation), and CRD (in relational-based knowledge distillation). We created a new benchmark that combines the experiments conducted by CRD and

Table 1: Test accuracy (%) of students and teachers with similar architectures for Hinton, Sem-CKD, CRD, and ModReduce. ↑ indicates outperforming Hinton's vanilla KD, whereas ↓ indicates underperforming

| Teacher | WRN-40-2 | WRN-40-2 | resnet56 | resnet110 | resnet110 | resnet32x4 | vgg13 |
|---------|----------|----------|----------|-----------|-----------|------------|-------|
| Student | WRN-16-2 | WRN-40-1 | resnet20 | resnet20 | resnet32 | resnet8x4 | vgg8 |
| Teacher | 75.61 | 75.61 | 72.34 | 74.31 | 74.31 | 79.42 | 74.64 |
| Student | 73.26 | 71.98 | 69.06 | 69.06 | 71.14 | 73.09 | 70.46 |
| Hinton | 75.39 | 74.21 | 71.70 | 70.99 | 73.66 | 74.32 | 73.62 |
| SemCKD | 75.10 (↓) | 73.11 (↓) | 70.91 (↓) | 70.95 (↓) | 73.47 (↓) | 75.55 (↑) | 74.08 (↑) |
| CRD | **76.12** (↑) | **74.91** (↑) | 71.72 (↑) | 71.35 (↑) | 73.65 (↓) | 74.97 (↑) | 74.39 (↑) |
| ModReduce | 75.44 (↑) | 74.84 (↑) | **71.99** (↑) | **72.01** (↑) | **74.34** (↑) | **75.78** (↑) | **74.64** (↑) |

Table 2: Test accuracy (%) of students and teachers with different architectures for Hinton, Sem-CKD, CRD, and ModReduce. ↑ indicates outperforming Hinton's vanilla KD, whereas ↓ indicates underperforming

| Teacher | vgg13 | resnet32x4 | WRN-40-2 | resnet32x4 | resnet32x4 | resnet32x4 | vgg13 | WRN-40-2 |
|---------|-------|------------|----------|------------|------------|------------|-------|----------|
| Student | MobileNetV2 | ShuffleNetV1 | ShuffleNetV1 | ShuffleNetV2 | vgg8 | vgg13 | ShuffleNetV2 | MobileNetV2 |
| Teacher | 74.64 | 79.42 | 75.61 | 79.42 | 79.42 | 79.42 | 74.64 | 75.61 |
| Student | 64.60 | 70.50 | 70.50 | 72.60 | 70.46 | 74.82 | 72.60 | 65.43 |
| Hinton | 68.72 | 74.59 | 75.45 | 75.73 | 72.48 | 77.21 | 75.89 | 69.02 |
| SemCKD | 68.66 (↓) | **77.21** (↑) | 76.93 (↑) | 78.07 (↑) | 75.02 (↑) | 79.14 (↑) | 76.24 (↑) | 69.77 (↑) |
| CRD | **69.66** (↑) | 75.77 (↑) | 76.59 (↑) | 76.57 (↑) | 73.68 (↑) | 77.71 (↑) | 76.26 (↑) | **70.13** (↑) |
| ModReduce | 69.23 (↑) | 76.96 (↑) | **77.14** (↑) | **78.23** (↑) | **75.21** (↑) | **79.51** (↑) | **76.76** (↑) | 69.37 (↑) |

SemCKD on CIFAR-100. Hence, our new benchmark contains 15 different combinations of teacher and student models architectures along with the accuracy of the expert teacher model, base student model, Hinton model accuracy, SemCKD model accuracy, CRD model accuracy, and our model "ModReduce" accuracy. The results collected in our new benchmark are as reported in tables 1 and 2, where the bold values represent the highest achieved accuracy. Our model surpasses the state-of-the-art benchmarks in 10 out of 15 experiments. It is worth noting that in the offline training of different students reported in tables 1 and 2, a weight for Hinton loss was added to both SemCKD and CRD students.

This was done to account for the fact that SemCKD reported their results with Hinton loss added to theirs in offline training. Furthermore, CRD reported training their student with CRD and Hinton that slightly improved upon CRD alone. Therefore, we preferred re-running all the experiments for Hinton, SemCKD + Hinton, and CRD + Hinton as they represent the offline knowledge distillation schemes with the best reported accuracies.

## 5 RESULTS AND DISCUSSION

In all the experiments, the 15 teacher-student models combinations shown in tables 1 and 2 were used to obtain the conclusions for our questions. To capture the improvement of our model over the existing knowledge distillation techniques, we utilized Average Relative Improvement (ARI); a metric that was first introduced by CRD (Tian et al., 2019) and was later used by SemCKD (Chen et al., 2021) in reporting their results. ARI provides a measure to test whether, on average, for the set of different architectures, ModReduce improved upon a certain knowledge distillation technique or not.

$$ARI = \frac{1}{M} \sum_{i=1}^{M} \frac{Acc^i_{ModReduce} - Acc^i_{KD}}{Acc^i_{KD} - Acc^i_{Stu}} * 100\% \qquad (3)$$

The first question was whether direct aggregation of offline losses introduced by Hinton, CRD, and SemCKD in a single loss function could improve upon using each loss function independently. For that, we ran an experiment that performs an aggregated distillation by adding the loss functions of Hinton, SemCKD, and CRD. Figure 3 shows that aggregating the three losses by simply adding the loss functions with their weights improves only upon the Hinton model. At the same time, it has equivalent performance to SemCKD and lower performance than CRD. From this experiment,

we can conclude that simply combining the loss functions of the different knowledge distillation techniques does not improve the accuracy of the student model.

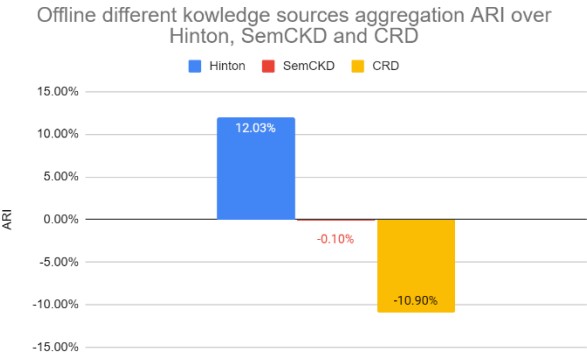

Figure 3: Average Relative Improvement of Combined Loss Function (Hinton + CRD + SemCKD) over each Method of them on its own

The next step was testing whether online aggregation of offline knowledge sources could improve the student's accuracy. We used four aggregation methods to create the teacher logits for the online learning step. Two of them were based upon the Peer Collaborative Learning (Wu & Gong, 2021) and On-the-Fly Native Ensembling (Zhu et al., 2018). The second aggregation method used a fully connected trainable layer to calculate the online teacher logits. Figure 4 shows a graph of the average relative improvement of the different aggregation methods used with ModReduce. ModReduce with ONE and Weighted Averaging (WAvg) have a positive ARI compared to training a student using any of the underlying single offline methods. Furthermore, using ModReduce with WAvg has the highest ARIs over all the underlying offline methods; those being $48.2\%, 25.50\%$, and $17.46\%$ over Hinton, SemCKD, and CRD, respectively.

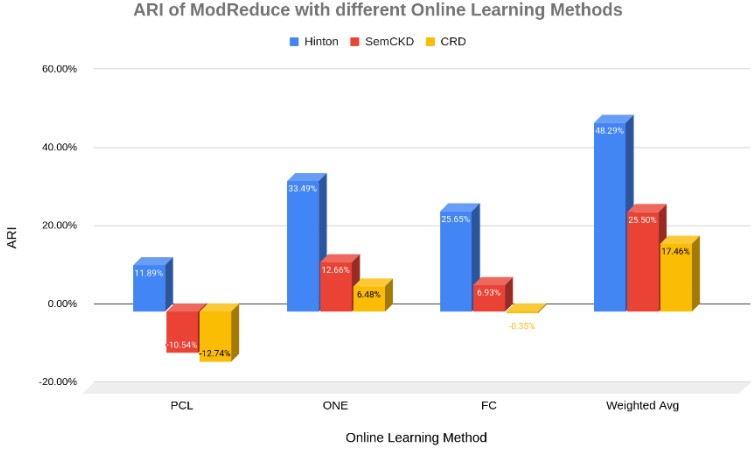

Figure 4: Average Relative Improvement (ARI) of ModReduce over SOTA Methods using Different Online Learning Techniques

Tables 1 and 2 show detailed results for the different experimentations. ModReduce, with WAvg as the aggregation method for the online learning step, is better in 10 out of the 15 experiments. While analyzing the five experiments in which ModReduce-WAvg is not the best performing, we observed that it also was never the worst performing compared to the other methods. For instance, in Figure 5a, despite trailing CRD, we notice that ModReduce-WAvg has an accuracy of $69.23\%$, improving up on SemCKD.

On the other hand, Figure 5b shows the opposite case where ModReduce-WAvg is trailing SemCKD. However, with an accuracy of 76.96%, it improves upon the state-of-the-art relational knowledge distillation (CRD).

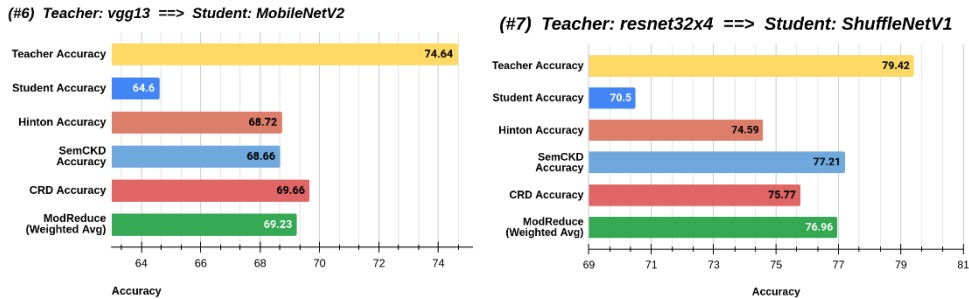

(a) VGG13 teacher and MobileNetV2 student    (b) Resnet32x4 teacher and ShuffleNetV1 student

Figure 5: Accuracy of student trained with ModReduce against Hinton, CRD, and SemCKD in experiments where ModReduce did not beat the Best performing method

These results, combined with the above ARI results, show that combining different offline knowledge students through online distillation generalizes better than a single offline knowledge distillation technique; thus, on average, ModReduce-WAvg produces better student models.

## 6    CONCLUSION

In this work, we introduced ModReduce: a novel multi-knowledge distillation with online learning framework. ModReduce aggregates knowledge distilled from three different sources: response-based, feature-based, and relational-based knowledge. This aggregation is performed via online learning between students, which also boosts their performance and enhances their generalization ability. Our experiments proved that using online learning as an aggregation method for different knowledge sources is better than combining the losses in a single student. We also showed that distilling three knowledge types is better than only using one or two types. Our results surpass the state-of-the-art SemCKD and CRD distillation schemes in 10 out of 15 experiments. More specifically, ModReduce outperforms SemCKD in 6 out of 7 experiments and outperforms CRD in 7 out of 11 experiments. Using the Average Relative Improvement (ARI) metric, ModReduce achieved 48.29% improvement over Hinton, 25.5% improvement over SemCKD, and 17.64% over CRD.

## 7    FUTURE WORK

Knowledge distillation is a prominent field with many research opportunities that could result in better performance and less costly architectures. Even though ModReduce has already achieved results that surpass the state-of-the-art benchmarks, a wide range of potential enhancements can be conducted.

We propose investigating the effect of switching from a synchronous offline-online training scheme to a sequential one (offline followed by online,) a potential enhancement to be tested. We have only replicated the experiments reported in SemCKD and CRD benchmarks in our work. However, other variations of teacher and student model architectures can be tested for further insights. Training student with different architectures, as the three students trained had the same architecture in any given experiment, might also be a good point to explore. Finally, we propose doing an extensive ablation study to see the effect of the different components of the system on the eventual result. This ablation study can test the effect of changing all variables or parameters one at a time.

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

# A APPENDIX

## A.1 DATASET

To accomplish the abovementioned experimentation rationale, ModReduce was tested on a series of classification tasks using the CIFAR-100 dataset, which consists of 100 classes containing 600 images each. CIFAR-100 has 500 training images and 100 testing images per class. The 100 classes are grouped into 20 superclasses. Each image comes with a "fine" label (the class to which it belongs) and a "coarse" label (the superclass to which it belongs). CIFAR-100 is the most used dataset in the state-of-the-art benchmarks we compared our results against, making it a convenient choice for our experimentations.

## A.2 PREPROCESSING

To have a fair comparison, we implemented an identical pre-processing to our two main benchmarks in Chen et al. (2021); Tian et al. (2019). All images are normalized by channel means and standard deviation. Moreover, some data augmentation is also implemented, such as random cropping and horizontal flipping.

## A.3 TRAINING DETAILS

Following our benchmarks, we use stochastic gradient descent with a Nesterov momentum of 0.9. We set the initial learning rate to 0.01 for MobileNetV2, ShuffleNetV1/V2, and 0.05 for other architectures. The number of training epochs is 240 for all models, and the learning rate is divided by 10 at epochs 150, 180, and 210. We set the mini-batch size to 64 and the weight decay to $5 * 10^{-4}$. The hyperparameter $\beta$ of SemCKD is set to 400, while the temperature T of Hinton KD is set to 4 throughout our experiments.

