# OpenReview forum: "ModReduce: A Multi-Knowledge Distillation Framework with Online Learning"
_ICLR.cc/2023/Conference — Submitted to ICLR 2023_

### Official Review · Reviewer_NDC7 · 2022-10-23

**Confidence:** 4
**Correctness:** 2
**Technical Novelty And Significance:** 2
**Empirical Novelty And Significance:** 2
**Recommendation:** 3

**Clarity, Quality, Novelty And Reproducibility:**

The clarity, the quality and reproducibility of the paper are ok. The quality of the presentation could be considerably improved (see above comments).

**Strength And Weaknesses:**

Strengths:

1. The addressed problem is interesting.
2. The general idea of the proposed approach is well motivated.
3. The observations with respect to the different knowledge distillation approaches are interesting.

Weaknesses:

1. Although the overall idea is well motivated, the single design decisions for the proposed approach seem to be ad-hoc (e.g., the steps of Algorithm 1 and the diagram of Figure 2 are not motivated). There are different ways to combine the three basic knowledge distillation approaches, why is your strategy the one to go for?
2. The evaluation on just one dataset is rather weak, and it is not clear whether the results generalize over other datasets.
3. The presentation can be improved. Many plots, diagrams, figures are too bulky for the conveyed message, many of them could be removed and replaced through single sentences in the text.



**Summary Of The Paper:**

The authors propose ModReduce, a framework to transfer the three knowledge types (knowledge on the produced output, knowledge on internal workings/intermediate layers, and knowledge on interactions between data points) from a teacher neural net model to a student neural net model by using a combination of offline and online knowledge distillation. The proposed approach is evaluated in comparison with the three mentioned types of knowledge distillation techniques on the CIFAR-100 dataset.

**Summary Of The Review:**

Although the overall idea is well motivated, the single design choices for the proposed approach seem to be ad-hoc (e.g., the steps of Algorithm 1 and the diagram of Figure 2 are not motivated). There are different ways to combine the three basic knowledge distillation approaches, and it is not clear why the proposed strategy the one to go for. The experiments are rather weak, and it is not clear how and whether they generalize over other datasets. The presentation can be improved. Many plots, diagrams, figures are too bulky for the conveyed message, many of them could be removed and replaced through single sentences in the text.

---

### Official Review · Reviewer_Ggs6 · 2022-10-25

**Confidence:** 3
**Correctness:** 3
**Technical Novelty And Significance:** 3
**Empirical Novelty And Significance:** 3
**Recommendation:** 5

**Clarity, Quality, Novelty And Reproducibility:**

Clarity: The paper nicely explains the different components of the system, being the individual (selected) offline knowledge distillation approaches and the online schemes to transfer knwledge.
Quality: The approach combines offline knowledge distillation schemes, which is a well-motivated direction. The conducted evaluation is sufficient wrt used architectures and datasets, but does not make clear if the approach can beat online approaches
Novelty: The approach builds on available components, but still provides a new perspective on combining them. The related work section introduces online/offline schemes and prominent approaches, but does emphasize where selected can be placed in terms of performance.
Reproducibility: The evaluation gives a good degree of details to reproduce the experiments, but no code is provided.

**Strength And Weaknesses:**

Strengths:
* The approach is well-motivated and is well-designed. The proposed architecture enables to reuse online schemes for updating individual students with specialized knowledge
* The empirical evaluation uses sensible architectures and tasks, and show results for the CIFAR dataset. The results show an improvement against individual offline knowledge distillation schemes

Weaknesses:
* The paper does not make clear how the paper compares against the global State-of-the-Art of knowledge distillation and only shows relative improvements. It would be important to discuss / show this in the paper. More specifically: Did you compare to SOTA for Online KD? The results improve wrt to offline KD, but the added value for KD approaches as a whole are not discussed in depth.
* A second point for the evaluation: It is also mentioned that two types of knowledge have been combined before, but this is not covered in the evaluation. How would your system perform against these?



**Summary Of The Paper:**

The paper proposes a novel two-stage knowledge distillation approach using a teacher network and multiple students which, in essence, combines offline response-based, feature-based and structural knowledge distillation using an online scheme. The authors do so by evaluating for online schemes, namely PCL, ONE, FC, and Weighted Averaging. The authors evaluate their approach on CIFAR100 for selected teacher and student architectures. The experiments show that the combination of the three offline schemes only works well when using an appropriate online scheme, but then can provide better results compared to individual offline knowledge distillation strategies.

**Summary Of The Review:**

The paper is proposes an intersting and well-motivated approach for combining available offline knowledge distillation approaches. The authors nicely make the case for the possible merit of combining respnse-based, feauture-based and relational knowledge and propose a sensible system to achieve this goal. The paper shows the relative improvements of the trained models compared to chosen offline distillation approaches, but I am lacking a clear result on the advancement of the knowledge distillation field. What are the consequences of the results? The authors state that the individual appraches (e.g. SemCKD and CRD) are the respective State-of-the-Art, but can the reader assume there are no other combinations / individual knowledge distillators that are better? Maybe this can easily be remedied in the related work section and clear statements in the results, but in the current version it is unclear to me.

---

### Official Review · Reviewer_5hfu · 2022-10-25

**Confidence:** 4
**Correctness:** 3
**Technical Novelty And Significance:** 2
**Empirical Novelty And Significance:** 2
**Recommendation:** 3

**Clarity, Quality, Novelty And Reproducibility:**

The paper is generally well organized and written in general. However, it is unclear why the distillation needs to be divided into two phases: an offline distillation followed by an online distillation. Although it seems easy to reproduce the results, the proposed framework is quite simple, and it just tries some combinations of existing distillation methods.

**Strength And Weaknesses:**

Strength:

(1) The proposed framework, named ModReduce, is simple and easy to implement.

(2) They show that combining three different distillation methods (response-based, feature-based, and relational-based knowledge) can improve the performance of student models

(3) ModReduce outperforms existing knowledge distillation methods in terms of Average Relative Improvement (ARI) on the CIFAR-100 dataset.

Weakness:

(1) Although the combination of three types of distillation methods has not been investigated, this study just combines existing distillation methods by simply adding some losses that were proposed in previous literature. The novelty seems a bit insufficient to me.

(2) As shown in Tables 1 and 2, the improvement of student models trained with ModReduce is not significant (less than 1%), compared to existing distillation methods.

(3) Only a dataset (i.e., CIFAR-100) was used to evaluate the proposed method.


**Summary Of The Paper:**

A framework called ModReduce was proposed to distillate knowledge from a teacher model by combining response-based, feature-based, and relation-based methods. The proposed distillation process is also divided into two phases: an offline distillation followed by an online distillation. The experimental results show that combing three different distillation methods (i.e., using three types of loss each defined for a method) yields better results than that only using one or two methods.

**Summary Of The Review:**

This work just combines existing distillation methods by simply using multiple losses that were proposed in previous literature at the same time, and tries some combinations of them. The novelty seems a bit insufficient. Besides, experimental results show a slight increase in accuracy on just one dataset.

---

### Official Review · Reviewer_9MXc · 2022-11-02

**Confidence:** 5
**Clarity, Quality, Novelty And Reproducibility:** n/a
**Correctness:** 2
**Technical Novelty And Significance:** 1
**Empirical Novelty And Significance:** Not applicable
**Recommendation:** 1

**Strength And Weaknesses:**

Very bad paper, no novelty.

**Summary Of The Paper:**

This paper proposes to transfer teacher knowledge in response, feature, and relation together.

**Summary Of The Review:**

n/a

---

### Decision · Program_Chairs · 2023-01-20

**Decision:**

Reject

**Justification For Why Not Higher Score:**

The lack of novelty does not allow the paper to be accepted at ICLR.

**Justification For Why Not Lower Score:**

N/A

**Metareview: Summary, Strengths And Weaknesses:**

The paper proposes an approach for knowledge distillation, which supports three types of knowledge (response-based, relational-based, and feature-based).

The reviewers identified the following strengths and weaknesses.

Strengths:
- Simple and easy-to-implement framework
- Positive evaluation results

Weaknesses:
- Insufficient novelty - it is only a combination of existing methods
- Not sufficiently taking related work into account
- Lack of SOTA algorithm comparisons; relative comparisons only performed on a single data set
- Several claims are not well supported